# A Practical Bioinformatics Workflow for Routine Analysis of Bacterial WGS Data

**DOI:** 10.3390/microorganisms10122364

**Published:** 2022-11-29

**Authors:** Aitor Atxaerandio-Landa, Ainhoa Arrieta-Gisasola, Lorena Laorden, Joseba Bikandi, Javier Garaizar, Irati Martinez-Malaxetxebarria, Ilargi Martinez-Ballesteros

**Affiliations:** 1MikroIker Research Group, Department of Immunology, Microbiology, and Parasitology, Faculty of Pharmacy, University of the Basque Country UPV/EHU, 01006 Vitoria-Gasteiz, Spain; 2Bioaraba, Microbiology, Infectious Disease, Antimicrobial Agents and Gene Therapy Group, 01009 Vitoria-Gasteiz, Spain

**Keywords:** foodborne pathogens, whole-genome sequencing, bioinformatics workflow, Galaxy

## Abstract

The use of whole-genome sequencing (WGS) for bacterial characterisation has increased substantially in the last decade. Its high throughput and decreasing cost have led to significant changes in outbreak investigations and surveillance of a wide variety of microbial pathogens. Despite the innumerable advantages of WGS, several drawbacks concerning data analysis and management, as well as a general lack of standardisation, hinder its integration in routine use. In this work, a bioinformatics workflow for (Illumina) WGS data is presented for bacterial characterisation including genome annotation, species identification, serotype prediction, antimicrobial resistance prediction, virulence-related genes and plasmid replicon detection, core-genome-based or single nucleotide polymorphism (SNP)-based phylogenetic clustering and sequence typing. Workflow was tested using a collection of 22 in-house sequences of *Salmonella enterica* isolates belonging to a local outbreak, coupled with a collection of 182 *Salmonella* genomes publicly available. No errors were reported during the execution period, and all genomes were analysed. The bioinformatics workflow can be tailored to other pathogens of interest and is freely available for academic and non-profit use as an uploadable file to the Galaxy platform.

## 1. Introduction

Throughout the last decade, the use of whole-genome sequencing (WGS) has become paramount for routine use at laboratories worldwide. Its decreasing cost, coupled with its high speed and throughput, have led to significant changes in the investigation and surveillance of outbreaks of foodborne illnesses caused by a wide variety of microbial pathogens [1]. The epidemiological characterisation of isolates using conventional methods such as pulsed-field gel electrophoresis (PFGE) or multiple-locus variable-number tandem repeat analysis (MLVA), considered gold standards for bacterial typing, requires several labour-intensive assays that can take days to complete [1]. In contrast, WGS provides an ‘all-in-one’ solution, providing all the information required for pathogen typing and characterisation, including the detection of genes of interest (e.g., antimicrobial resistance (AMR) genes and virulence factors), serotype prediction, plasmid replicon detection and sequence typing, among others, in a shorter amount of time (3–5 days). This is all achieved with unprecedented resolution and at a relatively low cost per sample, avoiding the need for multiple sequential laborious molecular assays as well.

In addition, WGS has enabled the development of novel typing methods such as core-genome multilocus sequence typing (cgMLST) that expand the standard seven-gene MLST by including hundreds of loci and whole-genome single nucleotide polymorphism (wgSNP) analysis, which provides much greater discriminatory power compared to conventional methods such as PFGE or MLVA. Thus, the resolution up to the nucleotide level enables pathogen comparison and clustering with unprecedented precision [2], making WGS the indisputable candidate to replace conventional typing methods. Thanks to these advantages, the use of WGS is becoming more widespread for pathogen typing in both outbreak and routine surveillance studies, with an increasing number of national reference centres (NRCs) and laboratories (NRLs) integrating it into their routine activities [3]. The added value of WGS in surveillance monitoring and outbreak cases for many microbial pathogens of interest in public health has been illustrated extensively [2,3,4,5].

Despite the innumerable advantages of using WGS, several drawbacks concerning data analysis, data management and exchange, and mainly the lack of validation, hinder its integration into routine use in NRCs and NRLs. For centres or laboratories in smaller or less-developed countries, the integration of WGS is particularly troublesome as they do not always have equal access to resources compared to public health agencies or large laboratories (WHO, Geneva, Switzerland, 2018). The data analysis in particular represents a bottleneck, becoming a serious hurdle to overcome because it typically consists of a stepwise process that is complex and tedious for non-experts [6]. The amount of software for analysing data is continuously growing, providing users with multiple choices based on their particular aims and interests. Several commercial software packages such as CLC Genomics Workbench (CLC Bio, Aarhus, Denmark), BioNumerics (Applied Maths, Sint-Martens-Laterm, Belgium) and Ridom SeqSphere+ (Ridom GmbH, Münster, Germany) ease the analysis of data by providing desktop applications. However, in this paper we leave commercial software packages aside as their associated costs usually become unaffordable for many laboratories; our focus is free and user-friendly alternatives for analysing WGS data.

An overview of data analysis tools was published by the ENGAGE consortium [7], with the aim of establishing the capability of WGS for genomic analysis in Europe. Attempts to integrate WGS into routine surveillance were also made in the INNUENDO project [8], whose aim was to develop a standardised cross-sectional framework for the application of bacterial WGS for the surveillance of foodborne pathogens that embraces governmental organisations, authorities and research institutes from the food, veterinary and human sectors in Europe. Unfortunately, most software for WGS data analysis runs on the Linux environment and requires a certain level of expertise in command-line skills, thereby hindering the analysis process for those who are not bioinformaticians. This problem can be managed by increasing the availability of web-based tools that allow non-experts to analyse their data without the need for command-line experience [1]. Currently, several web-based platforms are available for pathogen characterisation through WGS. Hosted at the Technical University of Denmark, the Center for Genomic Epidemiology (CGE) (https://www.genomicepidemiology.org/ (accessed on 14 September 2022)) provides a number of tools for performing serotyping [9], AMR characterisation [10], virulence gene detection [9], plasmid replicon detection [11], MLST and phylogenetic clustering, among other processes [12]. The Bacterial and Viral Bioinformatics Resource Center (BV-BRC) website (https://www.bv-brc.org/ (accessed on 14 September 2022)) provides a set of tools for WGS data analysis focused on pathogens. Other websites for WGS-based analysis of specific species exist, such as EnteroBase (https://enterobase.warwick.ac.uk/ (accessed on 14 September 2022)) [13], which provides solutions for analysing WGS data. PubMLST (https://pubmlst.org/ (accessed on 14 September 2022)) is another popular web-based platform that maintains databases with sequence typing information and schemas for a wide variety of pathogens, enabling the query of WGS data to be analysed. Although these resources are useful and user-friendly, they do have several drawbacks, as tools and databases must be combined manually, making the process difficult and tedious. In addition, database versions and tool parameters can alter the output, giving rise to different interpretations, and, thus, limiting the comparability, reproducibility and exchange of data between laboratories.

Another alternative is Galaxy, an open-source, community-driven, and web-based platform that provides a broad array of tools for analysing WGS data in a user-friendly interface [14]. Indeed, software and tools required for the analysis can be linked and processes executed sequentially by creating automated workflows. Several papers have reported the use of Galaxy for foodborne outbreak investigations and surveillance of a wide variety of microbial pathogens [15]. Consequently, Galaxy constitutes an appealing choice for analysing WGS data as it enables users to perform analyses easily, quickly and intuitively. In this paper, we present and evaluate a Galaxy bioinformatics workflow designed for the characterisation of foodborne pathogens focused on the study of an outbreak. This workflow was designed with the aim of routine utilisation by non-bioinformatician laboratory staff.

## 2. Materials and Methods

### 2.1. Bioinformatics Workflow

#### 2.1.1. Data Processing and Quality Control

An overview of the bioinformatics workflow is provided in Figure 1. The workflow supports all WGS data generated by means of the Illumina technology. To perform an all-in-one pre-processing of FastQ files, Fastp [16] is used with default settings. First, low-quality raw reads are removed if the average Q-score is >15, minimum length is <40, and number of N bases is >5. Afterwards, reads are trimmed at front and tail by removing all residues with a Q-score < 15; adapter sequences are automatically detected and removed, and the polyG tail in 3’ends, commonly seen in NovaSeq and NextSeq data, is detected and removed. Pre-trimming and post-trimming data-quality reports are obtained and merged into a single report by MultiQC [17]. Prior to the assembly, paired-end reads are taxonomically labelled by Kraken2 [18] with default parameters and the PlusPF database (last update 17 May 2021) to identify species and detect possible contamination. Kraken2 outputs are plotted by means of a Krona chart [19], enabling a visual check for contamination. Processed paired-end reads are de novo assembled with the SPAdes genome assembler [20] using the Shovill pipeline (https://github.com/tseemann/shovill (accessed on 20 May 2022)) with default settings. Assembly statistics such as N50 value and number of contigs are calculated with QUAST [21] using default settings. 

#### 2.1.2. Strain Genotyping, AMR and Plasmid Replicon Detection

Genotyping of assembled genomes is performed by Staramr [22]. The MLST scheme from PubMLST is applied for genotypic sequence typing, allocating MLST profiles to each assembled strain. For AMR detection, the ResFinder database [23] is used for alignment-based detection. Assembled genomes are aligned to the database using BLASTn and only hits with >90% identity and >60% target coverage are retained. Additionally, for the detection of plasmid replicons, the PlasmidFinder database [11] is applied. In accordance with default recommendations for plasmid replicon detection, minimum percentage identity for BLASTn is set at >95% instead of 90%, while target coverage percentage is maintained. Strain serotyping is performed by *Salmonella* In Silico Typing Resource (SISTR) [24] to obtain the antigenic profiles for each strain; this tool is useful only for *Salmonella* genome investigation.

#### 2.1.3. Virulence-Associated Gene Detection

Genes related to virulence (such as *invA*, *pefA* and *spvC* for the adhesion and invasion of epithelial cells or *spiC* for intracellular survival or host defence escape) are detected using ABRicate (https://github.com/tseemann/abricate (accessed on 3 June 2022)) by means of the Virulence Factor Database (VFDB) [25]. Only hits with >90% identity and >60% target coverage are retained. 

#### 2.1.4. Genome Annotation

Prokka [26] is used for the annotation of features of the draft genomes. Coding sequences (CDS) and ribosomal and transfer RNA genes are determined, among others. Several standard output files are created for further analysis.

#### 2.1.5. Phylogenetic Analysis

To perform the core-genome-based phylogeny, Prokka’s GFF output files are used for pan-genome creation by means of the Roary pipeline [27]. Parameters set for core genome determination are 95% of identity for protein matches and 99% of the isolates having the designated gene for it; that is the way to be considered a core gene. For each isolate, a fasta file containing the concatenated core gene´s sequences is created, and subsequently a multi-fasta alignment for all strains is performed for phylogenetic tree construction by IQ-TREE [28] using default settings. Thus, core genome-based phylogeny is constructed in order to enable the clustering of all isolates.

For SNP-based phylogeny, variant calling and filtering is performed by Snippy pipeline version 4.6.0 (https://github.com/tseemann/snippy (accessed on 16 June 2022)). SNPs are filtered by a minimum SNP quality of 25 and minimum mapping quality of 30, covered by at least 10 reads. Snippy outputs for every isolate are combined into a core SNP alignment, and subsequently non-standard sequence characters are removed. An SNP distance matrix is created by computing the distance in SNPs between all sequences in the aligned fasta file. Additionally, SNP phylogeny tree construction from the polished fasta alignment is performed by IQ-TREE [28] using default settings. CFSAN-FDA framework [29] is applied for interpreting the SNP analysis. When there are 20 or fewer SNPs and the phylogenetic analysis shows a monophyletic relationship with bootstrap support of 0.9 or higher, isolates within the same clade are considered closely related and are likely to have emerged from the same source, assuming that they share a common origin. In the same way, isolates presenting more than 100 SNPs are considered to be distantly related, assuming that they do not share the same origin.

#### 2.1.6. Implementation and Availability

The workflow is available at https://github.com/aatxaerandio/Galaxy_Workflow_for_Genomic_Analysis (accessed on 11 November 2022) as a downloadable file. Users can download and import it into their Galaxy platform for non-profit or academic use. Our workflow is compatible with data from any Illumina sequencing platform. Furthermore, it allows parameter adjustment and use of other bioinformatics assays for strain characterisation and AMR gene detection with alternative databases (e.g., ARG-ANNOT, CARD or NCBI). A summary of the steps and software used on the bioinformatics workflow is available in Appendix A.

### 2.2. Validation of the Workflow

#### 2.2.1. Validation Dataset

A collection of 22 *Salmonella* isolates recovered from food and stool samples derived from a local outbreak that occurred in 2019 were obtained from the Basque Government Public Health Laboratory (PHL_716—PHL_738). Procedures for the isolation and detection of *Salmonella* were conducted following the ISO 6579-1 protocol [30]. Of the total of 22 isolates, 12 were from stool, eight from eggs and two from food products (omelette) involved in the outbreak. Strain serotyping, phage typing and pulsed-field gel electrophoresis (PFGE) were conducted by the National Microbiology Centre (Institute of Health Carlos III, ISCIII, Spain) as performed routinely in suspicious outbreak cases.

Genomic DNA of the isolates was extracted by means of a NucleoSpin Tissue DNA purification kit (Macherey-Nagel, Duren, Germany) following manufacturer´s instructions from overnight cultures grown at 37 °C in tryptic soy broth (TSB) (Condalab, Madrid, Spain). Sequencing libraries were prepared using the MiSeq library preparation kit (Illumina, San Diego, CA, USA) and paired-end sequenced on an Illumina MiSeq instrument with a 150-bp paired-end protocol by the General Services of the University of the Basque Country UPV/EHU (SGIker). All WGS data generated for these samples have been deposited in the NCBI Sequence Read Archive (SRA) under accession number (PRJNA891285).

Based on the serotyping results obtained for the isolates described above, 182 *Salmonella* genomes from different countries, sources and time periods were downloaded from the European Nucleotide Archive (ENA) and NCBI repository and added for comparison and validation of the workflow. Therefore, the isolates dataset comprised 204 *Salmonella* genomes from numerous sources and different serovars (149 *Salmonella enterica* serovar Enteritidis, 31 *Salmonella enterica* serovar Bovismorbificans, 18 *Salmonella enterica* serovar Paratyphi B, and six *Salmonella enterica* subspecies *arizonae*).

#### 2.2.2. Validation Strategy

The designed workflow repeatability and reproducibility were evaluated by running the bioinformatics workflow on the same dataset twice locally, on the one hand, in the Galaxy environment as previously noted and, on the other, in two other computational environments. Those two computational environments were an Ubuntu 18.04.4 LTS (64-bit) server (8 GB RAM, 4 × Intel Core i7-8665U CPUs) and a Red Hat Enterprise Linux 7.9 (64-bit) server (32 GB RAM, 48 × Intel Xeon Silver 4116 CPUs). Each tool of the workflow was run in the same order on the command-line in both computational environments. Tool versions were equal in all the analyses performed. Results obtained from Galaxy and both computational environments were subsequently compared.

### 2.3. Further Genomic Analysis

As the isolates of this study belonged to the genus *Salmonella*, additional analyses were performed by means of external software, which is not part of the Galaxy workflow. The cgMLST profiles were obtained using cgMLST Finder 1.1 from the CGE (https://cge.cbs.dtu.dk/services/cgMLSTFinder/ (accessed on 14 June 2022)) based on the 3002 loci scheme developed by EnteroBase *Salmonella* database (https://enterobase.warwick.ac.uk/ (accessed on 14 June 2022)). Furthermore, Interactive Tree Of Life (iTOL) [31] was used for phylogenetic tree visualisation in both cases, the SNP and the core genome-based phylogenetic trees together with isolates metadata.

## 3. Results

### 3.1. Validation of the Workflow

The workflow was uploaded to the European Galaxy server (https://usegalaxy.eu/ (accessed on 16 July 2022)) and executed on the total number of isolates in the study. No errors were reported throughout the execution period. In parallel, the same dataset was analysed twice by running these tools one by one in two different computational environments. These validation analyses yielded the same outputs, showing 100% agreement between them.

### 3.2. Data processing and Quality Control

Read-trimming data and assembly statistics for all 204 genomes of the dataset are provided in Appendix A. The percentage of forward and reverse reads surviving was high (>90%) in the majority of samples except for two samples where percentages were 80.3% and 86.6%. The fraction of reads with a mean quality score > Q30 was high in the majority of samples and did not decrease from 81%. 

The N50 value, a metric used as a proxy for assembly quality, fluctuated punctually among samples. Although the vast majority of raw reads passed the trimming filtering, few genomes (IN_SeE_AU_084, IN_SeE_AU_104, IN_SeE_AU_151, ERR017993, SRR1060731, SRR1966201 and SRR3049093) showed a higher number of contigs (>100) in comparison with others. As a result, N50 values in those genomes are lower, indicating highly fractured genomes. Disregarding highly fractured genomes, average contig length was 32 for the total set of isolates. No significant evidence of contamination was detected in the assay.

### 3.3. Strain Characterisation

Results obtained from all the assays performed are summarised in Appendix A. Species identification and serovar prediction obtained by Kraken2 and SISTR were 100% in agreement with the dataset used in the study. Up to 13 different STs were identified among the total dataset; ST42 in *S. Paratyphi* B var. Java strains (n = 18), ST2811 in *S. arizonae* strains (n = 6), four STs for *S. Enteritidis* strains (ST11, n = 146; ST3233, n = 1; ST5094, n = 1; and ST5280, n = 1) and seven STs for *S. Bovismorbificans* (ST142, n = 8; ST1312, n = 8; ST1499, n = 7; ST150, n = 4; ST2345, n = 2; ST377, n = 1; ST2723, n = 1). Regarding cgMLST, up to 152 cgSTs were identified: five for *S. arizonae* (ST113232, n = 2; ST17246, ST42261, ST236629 and ST238128, one isolate per each cgST); 17 cgSTs for *S. Paratyphi* B var. Java (ST51845, n = 2; while one isolate was allocated for the remaining 16 cgSTs); 24 cgSTs for *S. Bovismorbificans* (ST274809, n = 8; while only one isolate was allocated for the remaining 23 cgSTs); and 106 cgSTs for *S. Enteritidis* (ST101570, n = 22; ST149200, n = 10; ST133290, n = 9; ST60723, ST133244, ST136918, ST136945 and ST136966 two isolates per each cgST; the remaining 98 cgSTs were allocated to one isolate each) (Appendix A).

AMR gene detection revealed up to 12 different predicted resistance profiles. The *aac(6’)-Iaa* gene, conferring predicted resistance to gentamicin, was detected in all the genomes except for *S. arizonae* isolates (no antimicrobial resistance genes were detected on those genomes). The majority of the isolates carried only the *aac(6’)-Iaa* gene, but a few (n = 11) carried up to eight AMR genes. Disclosed information about the predicted phenotype and genes detected in each strain is presented in Appendix A.

An overview of the detection of plasmid replicons is also shown in Appendix A. Plasmid replicons were detected in 61.3%, 96%, 50% and 5.5% of *S. Bovismorbificans*, *S. Enteritidis*, *S. arizonae* and *S. Paratyphi* B var. Java isolates, respectively. The most frequently identified plasmid replicons among the isolates were IncFII (74%) and IncFIB (71.6%), followed by Incl1 (8.3%) and IncX1 (4.9%). No plasmid replicons were detected in 18.6% of the isolates. Further data regarding identity percentage and coverage percentage, contig start and end position, and AMR and plasmid accession numbers for detected replicons are available in Appendix A.

Virulence-related genes were also detected in the analysis (Appendix A). A total of 125 virulence factors were identified in the entire dataset. Virulence profiles were in accordance with the *Salmonella* serotype; isolates belonging to the same serovar showed the same virulence-factor profiles despite slight differences. The numbers of virulence-related genes identified were 101 to 108 for *S. Enteritidis*, 91 to 111 for *S. Bovismorbificans*, 97 to 114 for *S. Paratyphi* B and 43 to 45 for *S. arizonae*. 

### 3.4. Phylogenetic Analysis

A total of 2516 genes were selected to compose the core genome of the *Salmonella* dataset. Core-genome-based phylogeny (Figure 2) allowed strain clustering up to serovar level and showed concordant results with those obtained with species identification and serovar prediction. 

Considering that SNP analysis is better suited for closely related isolates, a single analysis was performed within *S. Enteritidis* isolates (n = 149), using *S. enterica* subsp. *enterica* serovar Enteritidis str. P125109 (GenBank assembly accession GCA_000009505.1) as the reference genome. A total of 472 SNP positions were identified in the tested *S. Enteritidis* dataset. In this analysis, 99 isolates from the 149 studied were clustered into nine different clades (A, n = 6; B, n = 4; C, n = 6; D, n = 10; E, n = 10; F, n = 12; G, n = 15; H, n = 8; I, n = 28). The SNP matrix with the clades identified for *S. Enteritidis* genomes is available in Appendix A. Isolates from stool samples from the local salmonellosis outbreak that occurred in 2019 (in bold) were clustered with 15 other strains. Fewer than 20 SNPs were detected among them. 

## 4. Discussion

Foodborne pathogens are an important problem worldwide and are responsible for 600 million foodborne illnesses and 420,000 deaths annually. Bacterial pathogens such as *Salmonella*, *Campylobacter*, *Listeria*, *Escherichia* and *Brucella*, among others, remain the most hazardous due to the large numbers of cases they cause and the resulting economic burden [32]. Surveillance programs are responsible for the identification, outbreak control and tracking of foodborne pathogens, and the information gathered throughout an epidemiological investigation is essential for the evaluation of the effectiveness and for the development of new prevention strategies [4,33].

For decades, epidemiological investigations have been conducted through conventional methods such as MLVA and PFGE, the latter being considered the gold standard. Throughout the last decade, WGS has become increasingly available for routine use worldwide due to its unprecedented resolution, high speed and low cost per sample. WGS provides the bulk of information on bacterial isolates such as serotyping prediction, genotyping, detection of interesting genes (e.g., AMR and virulence-related genes), plasmid detection, and phylogenetic relationship determination, among others, in a much shorter time frame. Due to these characteristics, WGS is becoming highly useful and practical in epidemiological studies [2,4,5,33,34]. However, while the use of WGS implies considerable advantages over conventional methods, several drawbacks concerning data analysis, management and exchange, and a generalised lack of validation are hindering its widespread use. Currently, a broad array of tools is available for analysing WGS data published and available on the web. However, a certain level of expertise in command-line computing is required as most of the software is run in Linux environments. Thus, few alternatives are available that combine user-friendliness and all-in-one analysis at no cost. In that scenario, we present a Galaxy workflow for the characterisation of an important foodborne pathogen, *Salmonella*, coupled with a dataset as test data, with the aim of evaluating this tool for possible routine use by non-bioinformaticians.

As test data for the workflow validation, 204 *Salmonella* genomes were used: *S. Enteritidis* (n = 149), *S. Bovismorbificans* (n = 31), *S. Paratpyhi* B (n = 18) and *S. arizonae* (n = 6). Workflow design concerned the pre-processing of raw data and the general downstream genotyping assays for microbial pathogens, combined with serotyping, AMR gene and plasmid detection, virulence factors identification, core genome creation and phylogenetic clustering. Workflow was executed on the European Galaxy server (https://usegalaxy.eu/ (accessed on 16 July 2022)), and no errors were reported during the running time. For the evaluation of workflow repeatability and reproducibility, the bioinformatics workflow was run twice on the same dataset in the same and different computational environments. Each tool was run in the same order on the command line in both Ubuntu and Red Hat Enterprise Linux environments, and the results were 100% congruent with those obtained by the Galaxy server. All raw reads from isolate-sequencing data passed the trimming filtering, showing high-quality ratios. No evidence of contamination was detected in the dataset used, although a few samples displayed a higher number of contigs. Samples with a high number of contigs are deduced to be highly fragmented, due more to the sequencing of genomes than to possible errors in the assembling process.

The results obtained from the computational analysis showed that the designed pipeline is able to determine different characteristics among strains. In this case, 13 STs, 153 cgSTs, and a wide range of AMR-related genes and plasmid replicons as well as virulence gene profiles were identified. Predicted resistance to gentamicin was detected in the majority of the isolates of the study, and few of them carried *bla*_TEM-1B_, *bla*_TEM-1C_ or *bla*_SHV-2_ genes coding for extended-spectrum β-lactamases (ESBL) that are able to degrade a wide range of antibiotics such as the cephalosporins and become a food safety concern [35]. However, further considerations regarding results interpretations should be taken into account, as bioinformatics assays are performed at the genetic level, meaning that they may not correspond to real phenotypes [36]. Few plasmid replicons were identified in *S. Paratyphi* B and *S. arizonae* isolates, while numerous replicons were identified in the majority of *S. Enteritidis* and *S. Bovismorbificans* isolates. IncFIB and IncFII were the most abundant plasmid replicons identified in the dataset. Both plasmid replicons belong to the family of IncF, which is widely distributed and restricted to the Enterobacteriaceae family, in particular *Salmonella enterica*, *Shigella* and *Escherichia coli*, and carry virulence factors, AMR, cytotoxins, and adhesion factors [37]. Our results are similar to those of previously published studies [38,39].

Core-genome-based phylogeny was constructed in order to enable the clustering of all the isolates tested in this study. Isolates were clustered in accordance with their serotype and cgST, showing differences between serotypes or among different isolates within the same serotype. SNP analysis was performed only with *S. Enteritidis* isolates as it is better suited to related isolates belonging to the same species or serovar. A total of 472 SNPs positions were identified, and nine different clades were observed. Following the guidelines to interpret SNP analysis [29], clades were designated based on the SNP differences between genomes. Strains sharing fewer than 20 SNPs are considered closely related and as having a common source. Thus, the clade I composition is remarkable. The 12 *S. Enteritidis* isolates belonging to the local outbreak of 2019 clustered in that clade with another 10 recovered from Ireland and the UK (years 2017 to 2020) and with six other strains recovered from Austria in 2012. Although they are not exact matches with regard to the same time and geographic location, they are closely related, probably because they emerged from a common source. This could be the first step in deciphering a cross-country transmission, although further investigation is required. In addition, previous studies within other bacterial species have shown that SNPs and cgMLST are congruent [34], as both techniques are suitable candidates for outbreak and epidemiological investigations.

Furthermore, we stress that several web-based and command line-free alternatives exist for pathogen typing and visualisation. In this manuscript, we leave commercial software aside because the associated costs usually become unaffordable for many laboratories [40]. Therefore, we focused on free and user-friendly alternatives for analysing WGS data. In this sense, the iTOL tool was used for visualising phylogenetic clustering and data because of its user-friendly interface. Alternatives for data visualisation are available as MEGA X [41] or Phyloviz [42], among others, which perform and visualise phylogenetic clustering of isolates. Moreover, platforms such as PubMLST, Bacterial and Viral Bioinformatics Resource Center (BV-BRC), EnteroBase and CGE are powerful tools for pathogen typing that have been used extensively worldwide, but they do not provide a personal instance to run genomic analyses or an automated pipeline for performing all the desired analyses at once. Several pipelines such as Bactopia [43], AQUAMIS [44], ASA^3^P [45], rMAP [46], Nullarbor (https://github.com/tseemann/nullarbor (accessed on 10 November 2022)) and TORMES [47] perform automated analyses of bacterial genomes and provide users with a wide range of tools to customise the analysis to their needs. In fact, the results are presented in appropriate formats to facilitate their interpretation. However, installing and running the workflow remains a challenge as it is undertaken strictly on the command line. Workflows hosted on Galaxy instances partially solve installation issues by providing a ready-to-use platform for performing analyses. Indeed, several Galaxy workflows that are used for different purposes have been published in the last decade [48,49,50,51], but only a few of them address bacterial pathogens [52] and cover their complete characterisation. Thus, workflows hosted on Sciensano Galaxy (https://galaxy.sciensano.be/ (accessed on 10 November 2022)) for the characterisation of *Neisseria meningitidis*, Shiga toxin-producing *Escherichia coli* and *Mycobacterium tuberculosis* were published [1,6,53]. Those workflows are supported with specific databases and analyses according to each of the three species to be studied, making it easy to execute and visualise reports.

Bioinformatics tools hold great promise with regard to enhancing the productivity and discrimination power of current conventional methods and may even replace them. However, for WGS to be used broadly, it must be implemented in user-friendly software that facilitates the entire process [54].

Despite the low number of Galaxy workflows for bacterial characterisation, some papers have reported the use of Galaxy in foodborne outbreak investigations and the surveillance of a wide variety of microbial pathogens [15], making it suitable for epidemiological investigations. Although extra computational power to run analyses is not necessary, the limited storage (up to 250 GB but expandable to user’s requirements) becomes a hurdle as it hinders it use and implementation in NRC and NRLs, as the volume of data and computational needs exceed Galaxy’s limits. Our study is relevant as we provide guidelines to perform the first steps in the analysis of WGS data from scratch that are specifically addressed to non-bioinformaticians. Even though the proposed workflow was tested with a dataset composed of isolates of the genus *Salmonella*, workflow components and desired parameters could be adapted to users’ aims and tailored to other bacterial organisms. Our bioinformatics workflow is available at https://github.com/aatxaerandio/Galaxy_Workflow_for_Genomic_Analysis (accessed on 11 November 2022) as an uploadable file to a user’s Galaxy account and ready to use as an automated pipeline.

## Figures and Tables

**Figure 1 microorganisms-10-02364-f001:**
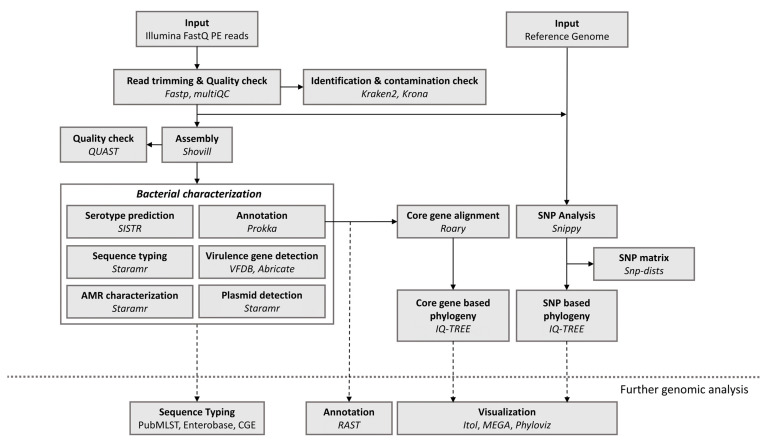
Overview of the bioinformatics workflow. Each box represents a component corresponding to a data analysis (indicated in bold). Software packages employed in each analysis are indicated in italics. Dashed lines indicate further genomic analysis available.

**Figure 2 microorganisms-10-02364-f002:**
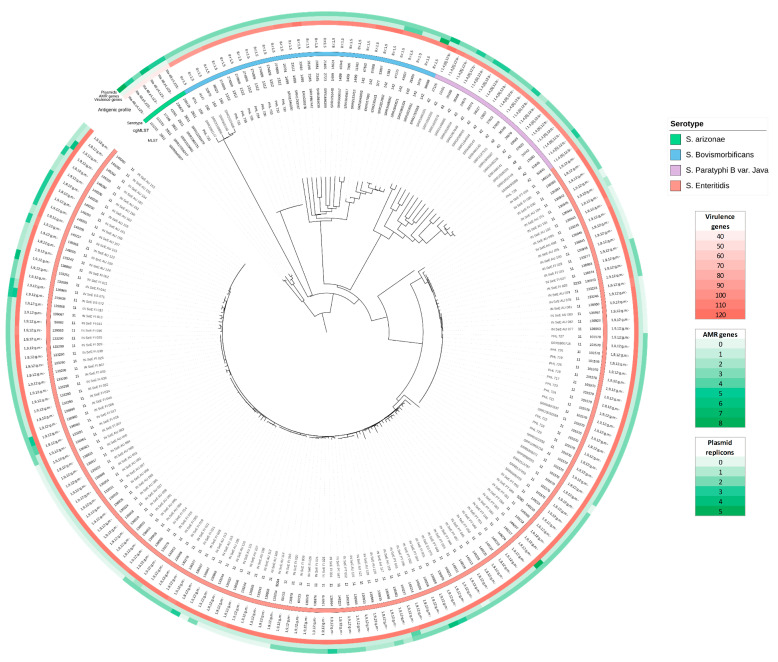
Core genome based minimum spanning tree containing an overview of the diversity contained in the tested dataset. The annotations are (from inner to outer rings): sample name, sequence type determined with the MLST and cgMLST scheme of Enterobase, sample serotype, antigenic profile, the number of virulence genes detected, the number of AMR genes detected and the number of plasmid replicons detected. The number of virulence genes, AMR genes and plasmid replicons are indicated according to the color legend. Full detailed information on the metadata for the characteristics of the isolates is available in Appendix A.

## Data Availability

The sequencing data from the study obtained after Illumina sequencing are available in the DDBJ/EMBL/GenBank databases under the accession number PRJNA891285. Other data presented in this study are available in Appendix A.

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
