# Peer review of "A Practical Bioinformatics Workflow for Routine Analysis of Bacterial WGS Data"

_microorganisms, 2022, doi:10.3390/microorganisms10122364_

Round 1

Reviewer 1 Report

The authors presented a workflow for WGS data analysis. They tested it using more than two hundred genomes. This work is well done and is practical for users.  However, the language need substantial editing to make the manuscript more clear. For example

Line 190-191,Line 228-230 unclear sentence

Line 256“ few” should be “a few”

Line 278 “are“ should be ”is"

The authors should go through the manuscript to correct similar questions

Reviewer 2 Report

The manuscript by Atxaerandio-Landa and colleagues aimed to develop a workflow for studying bacterial genomes using the widely known Galaxy platform. 

In general, the establishment of this workflow is important and serves not only laboratories without bioinformatics infrastructure, but also scientists who are unfamiliar with command-line tools and lack complex bioinformatics expertise. Despite the usefulness of the workflow, there are some important points that authors may want to consider: 

First, this workflow must be version controlled, and instead of providing the workflow as an additional file to the manuscript, I recommend authors to use some dedicated internet hosting services such as GitHub, GitLab, bitbucket or others. This allows to control the version of the workflow, to adopt future changes and developments, and to keep track of the workflow. Using such services will ensure that your workflow always remains alive and active. 

Second, there must be a detailed manual and instructions on how to use this workflow, preferably by creating a README file on the websites mentioned above. Even if there is a manual on the Galaxy itself, a step-by-step guide and tutorial for the user on your workflow would be helpful. 

Third, there was no validation of the selected tools by the authors. Most likely, the authors chose these software because they are widely available and frequently used. Right? Then maybe the authors should do a comparison between their workflow's tools hosted on Amazon and the command line tools to see if both approaches convey the same message. It would also be important to compare the features/components of this workflow with other pipelines already available and published.  

Also, all the mentioned tools of the workflow must be versioned and their license must be specified. In Figure 1, the authors mention abricate for virulence gene detection. I think they should add VFDB here in the figure. There is no mention of any tool for SNP-based phylogeny, snp matrix, or core genome-based phylogeny.  

Line 136 and line 147, I think the authors mean ">" the "greater than" sign and not "<" the "less than".

Line 151: is it "coding DNA sequences", "coding domain sequences" or just "coding sequences" "CDS"?

Round 2

Reviewer 2 Report

The authors replied to all my concerns and I have no further comments.